# Electrochemical Investigation of PEDOT:PSS/Graphene Aging in Artificial Sweat

**DOI:** 10.3390/polym16121706

**Published:** 2024-06-14

**Authors:** Boriana Tzaneva, Valentin Mateev, Bozhidar Stefanov, Mariya Aleksandrova, Ivo Iliev

**Affiliations:** 1Department of Chemistry, Faculty of Electrical Engineering and Technology, Technical University of Sofia, Kliment Ohridski Blvd., 8, 1000 Sofia, Bulgaria; b.stefanov@tu-sofia.bg; 2Department of Electrical Apparatus, Faculty of Electronic Engineering, Technical University of Sofia, Kliment Ohridski Blvd., 8, 1000 Sofia, Bulgaria; vmateev@tu-sofia.bg; 3Department of Microelectronics, Faculty of Electronic Engineering and Technology, Technical University of Sofia, Kliment Ohridski Blvd., 8, 1000 Sofia, Bulgaria; m_aleksandrova@tu-sofia.bg; 4Department of Electronics, Faculty of Electronic Engineering and Technology, Technical University of Sofia, Kliment Ohridski Blvd., 8, 1000 Sofia, Bulgaria; izi@tu-sofia.bg

**Keywords:** aging, conductive polymer, spray coating, artificial sweat, electrochemical stability

## Abstract

Herein, we investigate the potential application of a composite consisting of PEDOT:PSS/Graphene, deposited via spray coating on a flexible substrate, as an autonomous conducting film for applications in wearable biosensor devices. The stability of PEDOT:PSS/Graphene is assessed through electrochemical impedance spectroscopy (EIS), cyclic voltammetry (CV) and linear polarization (LP) during exposure to an artificial sweat electrolyte, while scanning electron microscopy (SEM) was employed to investigate the morphological changes in the layer following these. The results indicate that the layers exhibit predominant capacitive behavior in the potential range of −0.3 to 0.7 V vs. Ag/AgCl, with a cut-off frequency of approximately 1 kHz and retain 90% capacity after 500 cycles. Aging under exposure to air for 6 months leads only to a minor increase in impedance, demonstrating potential for storage under non-demanding conditions. However, prolonged exposure (>48 h) to the artificial sweat causes significant degradation, resulting in an impedance increase of over 1 order of magnitude. The observed degradation raises important considerations for the long-term viability of these layers in wearable biosensor applications, prompting the need for additional protective measures during prolonged use. These findings contribute to ongoing efforts to enhance the stability and reliability of conducting materials for biosensors in health care and biotechnology applications.

## 1. Introduction

The interest in conducting polymers for electrode fabrication on flexible substrates is driven by substantial advantages, including excellent mechanical properties and easy application through various techniques, including printing on textiles, paper, and flexible polymer substrates [1,2]. Some conductive polymers, such as poly(3,4-ethylenedioxythiophene):poly(styrenesulfonate) (PEDOT:PSS), offer additional benefits with proven biocompatibility, high electrochemical stability, high charge capacity, and commercial availability, making them highly suitable for in vitro, in vivo, and wearable sensors [3,4,5,6,7,8]. In these systems, the conjugated polymer PEDOT provides electrical conductivity, while PSS conveys ionic transport, contributing to an overall capacitive behavior comparable to that of conventional metal and carbon-based electrodes [9]. PEDOT:PSS is recognized as a leading organic electrode material, capable of replacing conventional carbon, metal (Au, Cu, Pt, etc.), and conductive glass-based electrodes, offering direct electron transfer at lower cost and process temperatures. Additionally, the intrinsic brittleness in metallic conductive layers, due to their high Young’s modulus, makes them unsuitable for flexible and stretchable electronics, and thus incompatible with soft biological tissues [10]. The fabrication of PEDOT:PSS films has been explored through various coating techniques, including dip coating, spin coating, and spray coating. For cost-effective application of large-area thin films over surfaces with a complex geometry, spray coating emerges as a viable option. This deposition technique is based on spraying the PEDOT:PSS solution onto a substrate, employing a nozzle at a precise distance and specific temperature [11]. While this route easily affords an uniform coating, some challenges related to surface wettability and substrate adhesion may arise. The technological and functional advantages of PEDOT:PSS-based layers have accelerated research on their properties over the last decade. The effects on morphology, mechanical properties, and conductivity of PEDOT:PSS for various deposition methods and conditions is often explored in the literature [12]. Charge transfer in intrinsically conducting polymers like PEDOT:PSS is likely influenced by both intra-grain and inter-grain structures due to their granular nature. The inter-grain structure is affected by grain elongation and stacking, primarily determined by the deposition conditions, especially temperature. The amount of the secondary dopant and its removal during the evaporation and annealing phases can impact the final alignment of grains across the film [13]. Favorable alignment of PEDOT:PSS grains, particularly minimizing the overall density of PSS regions along the charge pathway, contributes to the film conductivity. Consequently, smoother samples with horizontally aligned grains are expected to exhibit higher conductivity than the pristine sample at most temperatures.

Literature data on the electrochemical stability of these conductive layers, however, are not unequivocal. Some authors report a reduction in their charging ability with continuous cycling [14,15], while others suggest that PEDOT:PSS films remain stable over one hundred cycles [2,7]. Polymer materials are known to be susceptible to hydrolytic and oxidative processes attacking specific bonds in their structure. As a hydrophilic polymer, PSS is more sensitive to hydrolysis, as water molecules and ions can more freely penetrate its internal structure. Nevertheless, some long-term tests demonstrate remarkable stability of electropolymerized PEDOT:PSS films from 10^4^ to 10^9^ bipolar input pulses [8,16,17].

The long-term stability of the layers is directly related to their adhesion to the substrate. In most studies, conductive substrates such as metallic (Au, Pt, Ag), carbon, or ITO layers are used [2,18,19,20]. On some substrates, especially with thicker electrodeposition of PEDOT:PSS, cracking and delamination from the electrode surface have been observed due to weak adhesive forces and a lack of chemical interaction between them [2,18,19]. For instance, Mousavi et al. [2] demonstrated that PEDOT:PSS exhibits the highest adhesion to organic substrates with low conductivity and high roughness. Applying PEDOT:PSS coatings on flexible polymer substrates has also been achieved following their prior metallization [6] and various intermediate adhesion layers have been proposed to improve adhesion [8,17].

It is notable that electrochemical tests on PEDOT:PSS-based layers are usually conducted in neutral phosphate buffer saline (PBS) solution [2,8,16,17], and in some cases in media based on NaCl [12,21], KCl [22], or K_4_Fe(CN)_6_ solutions [23], albeit less commonly. Notably, tests in environments simulating human sweat are not found in the literature, despite the growing application of these polymers in wearable biosensors.

Conductive polymer-based electrodes are considered practical when they exhibit low and frequency-independent impedance over a broad frequency range [24], a characteristic that might be enhanced through PEDOT:PSS modification with various additives. E.g., Gold–silver core–shell (Au–Ag) nanoparticles have been incorporated into PEDOT:PSS layers for paraoxon-ethyl detection, demonstrating a synergistic effect resulting in higher conductivity and improved electrochemical stability [25]. However, this approach is complex and costly, prompting exploration of simpler composites as alternatives. PEDOT:PSS also acts as a surfactant for non-covalent functionalization of exfoliated graphene, which is employed for the preparation of hybrid inks with mixed conductivity [26]. The conjugated aromatic chains of PEDOT:PSS strongly anchor onto the graphene surface, due to π–π interactions, without disrupting the electronic structure of graphene [26]. Combining PEDOT:PSS with graphene allows for the modulation of electrical conductivity and material functionality. These hybrid conductive inks PEDOT:PSS/Graphene, can be applied conveniently through spray-coating method. The resulting films exhibit approximately three orders of magnitude lower resistance compared to pure PEDOT:PSS [27], while maintaining high transparency, chemical and thermal stability, stretchability, and low contact resistance with organic materials [26]. Additionally, these polymer inks enable precise and cost-effective deposition of conductive polymer layers on various substrates, including flexible polymer substrates without the need for prior metallization [1,27,28]. Most reported application procedures of PEDOT:PSS/Graphene films include solar cells and light-emitting diodes [29]. However, these structures lack well-defined electrochemical characterization for wearable biosensing devices. Simultaneously, PEDOT:PSS-based polymer layers exhibit pronounced hydrophilic properties, suggesting a significant impact of the surrounding environment on their electrochemical characteristics. The promising conductivity and stability of the PEDOT:PSS/Graphene composite make it suitable for formation of conductive patterns on dielectric substrates for biomedical applications. Clarifying both their electrochemical stability and resilience in a simulated environment of human sweat is crucial for incorporating conductive pattern made of PEDOT:PSS/Graphene layers into various wearable sensor structures, especially in electrochemical sensors.

This paper explores the potential application of a composite layer made of PEDOT:PSS/Graphene ink deposited through spray coating on polyethylene terephthalate (PET) as an independent conductive pattern material. Stability of PEDOT:PSS/Graphene is investigated through electrochemical impedance spectroscopy (EIS), linear polarization (LP) and cyclic voltammetry (CV) during aging in ambient conditions and exposure to artificial sweat electrolyte for nine days. Morphological changes in the layer post-tests are examined using scanning electron microscopy (SEM). This study is a continuation of our previous work [30] where we elucidated the influence of the spray deposition modes (spray pressure, temperature, and number of passes) of PEDOT:PSS/Graphene ink on the thickness, sheet resistance, and roughness of the layers as well as on its electrochemical behavior in the artificial sweat. Here, we emphasize the aging and the long-term stability of the hydrophilic PEDOT:PSS/Graphene composite ink, choosing the layer with the smallest thickness (of 1 μm). Electrochemical studies of PEDOT:PSS/Graphene on insulating PET substrate provide advantages by avoiding interference at the polymer/metal interface and potential issues like electrolyte penetration, delamination, and noise introduction from this interface to the useful signal. To address flexibility and substrate insulation issues, we propose a test structure allowing electrochemical analysis of conducting polymer electrodes using conventional electrochemical techniques. Additionally, assessing the stability of polymer layers upon exposure to artificial sweat and extended air storage contributes to defining the applicability boundaries of PEDOT:PSS/Graphene layers as electrodes for routine applications.

## 2. Materials and Methods

### 2.1. Materials

The Graphene/PEDOT:PSS hybrid ink used in the present study was purchased as a finished product from Sigma Aldrich—Merck KGaA (Darmstadt, Germany). According to the production certificate, the conductive ink is prepared by dispersing 0.2 mg mL^−1^ PEDOT:PSS and 1 mg mL^−1^ electrochemically exfoliated graphene in dimethylformamide (DMF). Sodium chloride (NaCl, ≥99.0%, CAS №: 7647-14-5), ammonium chloride (NH_4_Cl, ≥99.5%, CAS №: 12125-02-9), acetic acid (≥99.8%, CAS №: 64-19-7), *D*,*L*-lactic acid (CAS №: 50-21-5) were used for artificial sweat preparation and were purchased from AlfaAesar (Karlsruhe, Germany).

### 2.2. Fabrication

The PET flexible (25 × 25 mm squares, 270 μm thick), underwent a cleaning process by sonication in isopropyl alcohol, followed by UV–Ozone surface treatment (λ = 265 nm, P = 350 W). Samples were fabricated employing a spray coating setup HS-AS18CK Haosheng (Ningbo, China) equipped with a regulating nozzle (HS-30 0.01–0.1 mm diameter range) and a heated substrate holder. The substrate temperature was maintained at 100 °C. The layers were deposited with an aerosol pressure of 3.5 mbar at a constant nozzle-to-substrate distance of 12 cm and 5 spray passes. The pre-heated substrate was exposed to the aerosol flow in 5-second cycles, allowing solvent evaporation and preventing material leakage. 

### 2.3. Material Characterization

Raman spectra of PEDOT:PSS/Graphene ink were obtained on a TO-ERS-532 spectrometer (Thunder Optics, Montpellier, France), based on a 532 nm laser source and a Raman probe, equipped with a ×20 microscope objective lens. Approx. 50 μL of the ink suspension were drop-casted on a glass substrate and dried at 80 °C for 15 min to remove the solvent. The recorded spectrum is presented in Figure 1. In the Raman spectrum of nanocomposite ink, the characteristic D, G and 2D peaks of graphene are seen at 1350, 1588 cm^–1^ and 2700 cm^–1^, respectively [27,31]. The peaks at ~1440 cm^–1^ and ~1500 cm^–1^ can be assigned to C=C symmetric and C=C asymmetric stretch in PEDOT:PSS, respectively [27,31,32]. The Raman spectrum of ink dropped on PET is a hybrid spectrum on which both the peaks of the ink and those of the PET substrate can be observed (Appendix A), while the spectrum collected on tested sample with a thin PEDOT:PSS/Graphene layer sprayed on PET presents only the strong peaks of PET substrate (Appendix A).

The sprayed PEDOT:PSS/Graphene film achieves thickness of 1.00 ± 0.05 μm measured by Alpha Step 100 (KLA-Tencor GmbH, Dresden, Germany). The obtained composite layers were found to have a sheet resistance of 111.6 ± 6 Ω sq^−1^, as estimated by Veeco FPP-5000 4 Point Prober (Veeco Instruments Inc., Plainview, NY, USA) and roughness of 9.2 ± 0.7 nm, as estimated by atomic force microscopy (Nanosurf, FlexAFM, Liestal, Switzerland). Scanning electron microscopy (SEM) analysis was performed on a Hitachi TM4000 (Hitachi, Tokyo, Japan) table-top microscope at accelerating voltage of 15 kV in the back-scattered electrons (BSE) mode and by Tescan LYRA (Brno, Czech Republic) at accelerating voltage of 20 kV.

### 2.4. Electrochemical Measurements

Electrochemical impedance spectroscopy (EIS), linear polarization (LP), and cyclic voltammetry (CV) were employed to investigate the electrochemical stability and aging in artificial sweat. A potentiostat–galvanostat PalmSens4 (PalmSens BV, Houten, Netherlands) equipped with a frequency response analyzer and PSTrace 5.9 software were used. For electrochemical tests, a model artificial sweat solution with a composition of 20 g L^−1^ NaCl, 17.5 g L^−1^ NH_4_Cl, 5 g L^−1^ acetic acid, and 15 g L^−1^ DL-lactic acid was used [33,34]. A fresh electrolyte was prepared before each test from stock solutions of the individual components, and the pH was adjusted to 4.7 using NaOH. The stock solutions were stored at a temperature of 5 °C. 

The electrochemical tests were conducted at a temperature of 25 °C in a three-electrode cell, as shown in Figure 2. The working electrode was a sample with a nanocomposite PEDOT:PSS/Graphene layer deposited on PET with an exposed area of 0.1 cm^2^. A Pt-plate (1 cm^2^) and Ag/AgCl electrode (3.0 M KCl, 0.21 V vs. SHE) were used as the counter and reference electrodes, respectively. All potentials are presented versus the Ag/AgCl electrode. EIS tests were performed at an open-circuit potential with an amplitude of 10 mV in a frequency range from 100 kHz to 10 mHz. The linear polarization (LP) was used for the electrochemical loading of the composite layer during aging tests in the electrolyte from −0.15 V to 1.0 V vs. Ag/AgCl at a scan rate of 1 mV s^−1^. CV with scan rates from 0.01 to 1.0 V was employed to determine the capacity of the tested layer. A total of 500 cycles from −0.3 to 0.7 V with a potential scan rate of 0.1 V s^−1^ were conducted to investigate the electrochemical stability of layer.

In the employed electrochemical cell design (Figure 2), the conductive polymer layer contacts a metal current collector from the front side, connected to the potentiostat. The current collector was fabricated from aluminum foil (38 μm thickness and 99.99% Al purity) with a 5 mm diameter opening, uniformly positioned around a silicone seal (1 mm gap), determining the path of current lines and the main resistance for charge transfer from the electrolyte/polymer boundary to the potentiostat. The voltametric characteristic of the aluminum-conductive polymer contact in the range from −1 to 1 V is linear, exhibiting a 7.7 mA V^−1^ slope in both potential scanning directions, thus indicating a pure ohmic contact between the two conductors. The small thickness of the aluminum foil allows its maximum approach to the electrolyte without hindering sealing with the silicone seal. Nevertheless, the 1 mm distance between the current collector and the electrolyte chamber can be considered sufficiently large to exclude the possibility of the resultant electrochemical signal being influenced by electrolyte penetration through the polymer to aluminum boundary and deterioration of the contact at this boundary, a phenomenon encountered in other arrangements of the current collector [24]. This would lead to liquid seepage between the insulating layers and internal corrosion [24,35] and the resulting decrease in impedance might skew the interpretation of the electrochemical data.

Under the above conditions, each of the 3 PEDOT:PSS/Graphene samples was tested at least in 3 points in different areas of the surface. The samples regarded as “fresh” were tested within a short timeframe (<1 week) after their deposition. The influence of two types of aging on the electrochemical performance of the “fresh” samples was tested: (1) “aged in air”, which were stored under exposure to air (50 ± 5% RH and 25 ± 1 °C) for 6 months, prior testing; and (2) “aged in artificial sweat”, which were tested under exposure to a simulated sweat liquid-phase environment, for up to 9 days.

## 3. Results

### 3.1. Electrochemical Stability

Cyclic voltammetry was employed to investigate charge transfer at the electrode/electrolyte interface. In the potential range from −0.3 to 0.6 V, characteristic reaction peaks were not registered either in the anodic or in the cathodic branch of the CV dependences (Figure 3a). Therefore, CV profiles suggest predominantly capacitive charging of the layer, and no faradaic reactions are observed [2]. The capacitive current linearly increases with the rise in potential scan rate from 0.01 to 1.0 V (Figure 3b). The slope of the linear fit to the dependence of the capacitive current density *j*_c_ (mA m^−2^) on the scan rate *ν* (V s^−1^) represents the voltametric capacitance (*C*_CV_) of the PEDOT:PSS/Graphene layer in artificial sweat [2,36]. The determined *C*_CV_ value stands at 111.6 mF m^−2^.

Assessment of the charging ability during continuous cycling (500 cycles) was conducted by calculating the voltametric charge, *Q_CV_*, from the hysteresis area of the CV curves (Appendix A) using the equation:(1)$$Q_{CV}=\frac{1}{2v}\int_{E_{1}}^{E_{2}}j(E)dE$$
where *ν* is the scan rate (0.1 V s^−1^), *E*_1_ and *E*_2_ are the potential window, and *j* is the current density at each potential. Figure 4 present the calculated values, according to Equation (1) and the capacity retention calculated as ratio of hysteresis area of a cycle *n* to that of the first cycle. The average voltametric charge values decrease from 12.4 to 10.0 μC cm^−2^, indicating a capacity retention after prolonged cycling of approximately 90.5 ± 5.1%. The degradation of the layer is most pronounced in the initial 10 cycles (approximately 3.5%).

The impact of electrochemical cycling on the impedance of the layer was investigated through EIS tests. Figure 5 depicts Nyquist and Bode plots of tests before and after 500 scans in artificial sweat in a 1 V potential window (from −0.3 to 0.7 V). In both tests, the Nyquist plot (Figure 5a) takes the form of an incomplete capacitive semicircle. After electrochemical cycling, the dependencies at high frequencies overlap, with an observed increase in the slope of the arc at low frequencies, reflecting an increase in the layer’s capacitance. The Bode plots presented in Figure 5b also show a slight increase in capacitive behavior after cycling at frequencies below 10 Hz, demonstrated by the higher phase shift. However, the observed differences can be considered negligible.

Low and frequency-independent impedance for PEDOT:PSS/Graphene layers is registered in the high-frequency range, where the PEDOT:PSS/Graphene layer exhibits a purely resistive character (the phase shift does not exceed −10°) and remains unchanged in magnitude after 500 cycles in artificial sweat. The cut-off frequency (*f*_cut-off_) for the transition from resistive to capacitive behavior slightly decreases from 2.53 to 1.63 kHz after cycling. These values are close to 1 kHz, which is a representative frequency in biosignals [2]. At low and medium frequencies, the impedance increases linearly, indicating the dominance of capacitive behavior in the layers. In this range, the magnitude dependence after electrochemical aging lies just below that of a freshly deposited layer and demonstrates a slight decrease in impedance (from 7.3 MΩ to 5.7 MΩ at 10^−2^ Hz) and an increase in the capacitive behavior of the layer (additional phase shift of approximately −7° at 10^−2^ Hz). The layer exhibits stable capacitive behavior in the frequency range of 0.1–300 Hz, as the phase angle remains more negative than −75°. The reduction in phase shift at low frequencies to −61° for the fresh layer is associated with parallel surface reactions contributing to charge transfer [37]. Presumably, during electrochemical cycling, some functional groups in PEDOT:PSS exposed to the electrolyte are deactivated, reducing the charge exchange, i.e., the resistive component of impedance. This aligns with the CV results discussed earlier.

Various equivalent circuits have been proposed for the analysis of PEDOT:PSS-based layers, ranging from the simplest serial RC circuit [21,38], Randles circuit, *R*_s_(*C*_dl_(*R*_ct_*W*)) [2,7,39], to significantly more complex ones involving multiple RC blocks and Warburg impedance (*W*) [2,8]. The choice of an equivalent circuit is significantly influenced by the substrates used and additives to the conducting polymer. We experimented with different equivalent circuits to analyze our results for PEDOT:PSS/Graphene on a PET insulating substrate. The best fit was achieved with the equivalent circuit denoted by the description code [*R*_s_([*R*_ct_(*R*_i_*Q*_i_)]*C*_dl_)], represented as an inset in Figure 5a. In this circuit, *R_s_*, *R_ct_*, and *R_i_* represent the solution resistance, charge transfer resistance of the external solid/liquid interface, and the internal resistance of the composite layer, respectively. The *C_dl_* present the contribution of the double layer capacitance to impedance. The (*R*_i_*Q*_i_) block connected in series with *R*_ct_ describes various transfer mechanisms within the interior of the PEDOT:PSS/Graphene composite, such as ionic flow in the PSS ion channels and electrical conductivity in graphene and PEDOT. Using a constant-phase element (*Q*_i_) reflects the surface inhomogeneity and porosity of the composite layer, resulting in a phase shift that does not reach the typical −90° for pure capacitance. The discrepancies between experimental behavior and combinations of purely electrical elements require the introduction of non-electrical elements such as *Q* into the equivalent electrical circuit to account for film properties, charge carrier mobility, conduction path, and driving forces [40]. Using a constant-phase element is especially necessary in cases of conducting polymers characterized by heterogeneity and disorder (at least at the molecular level). In the literature, this equivalent electrical circuit is used to describe the electrochemical behavior of organic layers with bipolar molecules that form a multilayered assembly, blocking the transfer of electron charge [41].

The fits achieved with the proposed equivalent electrical circuit are represented as black lines on the Nyquist and Bode plots (Figure 5), and the values of individual elements are given in Table 1. The χ^2^ values of both samples were obtained in the range between 10^−4^ and 10^−3^, indicating a sufficiently good fit. The most significant changes after 500 cycles are observed in the values of *R_ct_* and *C_dl_*, which increase by approximately two times, from 100.6 to 207.7 Ω and from 257.6 to 450.6 nF, respectively. The capacitance could increase either due to an increase in the active electrochemical surface of the electrode or the volumetric capacitance of the PEDOT:PSS layer. Both pathways are associated with changes in surface structure during electrochemical cycling [42].

The changes in the surface morphology of the conductive polymer layer were observed through scanning electron microscopy. SEM images in the back-scattered electron (BSE) mode, taken before and after 500 cycles in artificial sweat are presented in Figure 6. The BSE mode was chosen because the images are more contrasted, especially after aging of the layer (Appendix A). The image of the untreated surface shows relatively evenly distributed bright globular aggregates of PEDOT molecules linked with PSS. Figure 6b depicts a more heterogeneous area of the PEDOT:PSS/Graphene layer’s surface after electrochemical aging. In the BSE mode, the surface appears more heterogeneous with bright islands. The observed changes are likely a result of swelling and partial dissolution of the PSS matrix, while the distribution and size of PEDOT aggregates appear unchanged.

### 3.2. Air Aging

An essential characteristic of materials used in sensor devices is their stability during storage. The physicochemical interaction of organic layers with the surrounding environment is inevitable. It manifests in aging processes within polymeric macromolecules, the adsorption of oxygen, moisture, and contaminants, resulting in changes in the volume and surface properties of composite layers. The occurring changes inevitably impact the impedance of the layers and can, therefore, be adequately traced through electrochemical impedance spectroscopy. Figure 7a illustrates the impedance characteristics obtained from fresh PEDOT:PSS/Graphene layers and after exposure to air (RH approximately 50% and room temperature) for 6 months. No significant alteration in the Nyquist and Bode plots’ shape is observed, with partial overlap in the dependencies before and after air aging. The equivalent electrical circuit [*R*_s_([*R*_ct_(*R*_i_*Q*_i_)]*C*_dl_)] used for fresh layers is also applicable to air-aged layers. The values of the elements from this circuit are presented in Table 1.

The primary trends during air aging include a slight increase in the capacitive semicircle radius in the Nyquist plots, reflecting a mild elevation in the values of *R_ct_*, *R_i_*, and *C_dl_*. These results indicate that PEDOT:PSS/Graphene layers exhibit a weak tendency to age and increase their impedance when exposed to ambient air. The aging processes may be associated with drying and cracking of the layer over extended exposure periods [43].

### 3.3. Aging in Artificial Sweat Simultaneously with Electrochemical Loading

In addition to stability during sensor storage, it is crucial to determine the stability of the conducting material in operational conditions. In this context, the impedance of a PEDOT:PSS/Graphene layer was periodically measured during prolonged exposure to artificial sweat. Electrochemical impedance spectroscopy (EIS) tests were alternated with electrochemical loading of the layer through linear polarization within a wide potential range from −0.2 to 1.0 V. Anodic polarization predominates since oxidative processes are expected to exert a more pronounced negative influence on polymer aging processes. The results of impedance tests and polarization dependencies are presented in Figure 8 and Figure 9, respectively. Unlike air exposure, aging under operational conditions significantly alters the course of impedance characteristics. 

Notably, Nyquist and Bode plots for the layer after 24 h of exposure (red inverted triangles) are positioned close to those of the initial scan, maintaining stable capacitive behavior at frequencies below 1 kHz. After 48 h, substantial changes in the frequency dependencies of impedance magnitude and phase shift become evident. A significant increase in impedance magnitude is observed across the entire frequency range during exposure exceeding 24 h (Figure 8b). Higher impedance values indicate slower ion exchange and a reduced layer capacitance compared to the fresh state [44]. This is further emphasized in Figure 8c, where a decrease in phase shift (below −70°) is registered in the low- and mid-frequency range, accompanied by the formation of two peaks (at frequencies of approximately 1 kHz and 0.1 Hz). These changes in Bode plots demonstrate an increase in the resistive component’s contribution to the overall system impedance, possibly due to thinning of the conducting layer [2] and the emergence of pores or new phase boundaries with the electrolyte [41]. Furthermore, by the third day, the cut-off frequency (*f*_cut-off_) shifts to higher frequencies, indicating the deterioration in the layer’s quality. Considering that higher *f*_cut-off_ values are characteristic of thinner layers with the same material, it can be assumed that the occurring deteriorations during exposure to artificial sweat lead to layer thinning.

The prolonged contact with the electrolyte (beyond 2 days) leads to a decrease in the impedance magnitude in the mid- and low-frequency ranges (Figure 8b). This indicates that changes in the composite are likely associated with an increase in ion content within the interior of the polymer layer. Simultaneously a reduction in the capacitive charging of the layer is observed.

The deviation of the phase shift from −90° necessitates a change in the equivalent circuit. In the proposed circuit for the fresh layer, *C*_dl_ is replaced with a constant-phase element *Q*_dl_. The resultant fits from the equivalent circuit [*R*_s_([*R*_ct_(*R*_i_*Q*_i_)]*Q*_dl_)] are represented as black lines on the plots in Figure 8, and the values of the elements are presented in Table 2.

Contact with artificial sweat significantly elevates the values of all resistances in the equivalent circuit, with the internal resistance *R_i_* being the most affected. Structural and chemical changes in the layer also lead to a decrease in capacitances, with *Q_i_* values decreasing by approximately 1 order of magnitude, and those of *Q_dl_* by more than 3 orders of magnitude. The values of the frequency dispersion coefficient *n*_dl_ also decrease, dropping below 0.7 after 1 week.

Fitting some results with the proposed equivalent circuit is not sufficiently good, and the χ^2^ values are not representative enough. Achieving a complete coincidence between experimental results and fit of the equivalent circuit is a serious challenge for PEDOT:PSS/Graphene since this material encompasses phases with different conductivity mechanisms, further complicating modeling due to different capacitive effects manifesting in various frequency ranges [40]. Indeed, better fitting of experimental results of tests over two days in artificial sweat is achieved using more complex equivalent electrical circuits such as hybrid series/parallel models, involving more elements corresponding to distributed electronic phases in an ion-conductive environment [45]. Nevertheless, we judged that using the same circuit for all electrolyte aging results has more advantages in tracking degradation processes than reducing χ^2^ values at the expense of employing different and more complex equivalent electrical circuits in the modeling.

Between each EIS test, the sample is polarized in the range of −0.2 V to 1.0 V vs. Ag/AgCl at a rate of 1 mV s^−1^ for the purpose of electrochemical loading of the polymer layer. The obtained polarization dependencies are presented in Figure 9.

EIS spectra recorded before and after polarization of the layer from −0.2 to 1 V overlap completely, indicating that polarization, even at these slow potential scanning rates over a relatively wide potential range, does not significantly influence the properties of the composite, unlike the duration of its exposure to artificial sweat.

From the polarization dependencies, the exchange current density (*j*_0_) and polarization resistance (*R_p_*) were determined and are presented in Table 3. The results show a very low exchange current (around and below nanoamperes) and a very high polarization resistance (around and above 100 MΩ), which is expected for a layer that does not spontaneously interact with the surrounding electrolyte. The exchange current density (*j*_0_) of the electrochemical system depends on various factors of the material and the environment such as ion concentration, temperature, catalytic properties, and the specific surface area of the electrode [46].

After one day of exposure, the exchange current decreases to approximately 3 nA cm^−2^ and remains around this value until the end of the experiment. The lower exchange current indicates an increase in the resistance of the conducting layer, most likely due to the destruction and thinning (reduction in effective conductive cross-section) of the layer. After 4 days of exposure to artificial sweat, the potential shifts in the negative direction. This could be interpreted as an indication that the distance between the electrolyte and the aluminum current collector has decreased to the point where the ion flow reaches the aluminum surface, and the aluminum/conductive polymer interface begins to influence the recorded polarization current signal.

SEM observations of the sample after prolonged exposure to artificial sweat show the loss of the PSS phase, exposure of PEDOT:PSS clusters (Figure 10a, bottom left corner, Figure 10b, Appendix A), with extensive areas in the middle of the test field completely destroyed (Figure 10a, bottom right corner). In certain areas, the destruction of the composite extends beyond the seal line (Figure 10a, yellow dashed line), primarily through cracks in the layer.

Unfortunately, although clearly visible by SEM layer destruction, our attempts to determine with Fourier-transform infrared spectroscopy (FTIR) the chemical changes in the polymer layer were unsuccessful. The obtained FTIR spectra are dominated by the peaks of the PET substrate (Appendix A) and those characteristic of S–phenyl (at 1157, 1121 and 1012 cm^−1^), C=C bonds of the thiophene ring (at 1532 and 1521 cm^−1^), C–C bonds of the thiophene ring (at 1356 and 1312 cm^−1^) and for the sulfonic acid group of the PSS (at 1196 cm^−1^) are not clearly registered [23].

## 4. Discussion

A specific feature of organic layers is their loose structure and the possibility of penetration by various components from the surrounding environment. In this regard, PEDOT is considered a dense material, taking several days to wet with electrolyte [45]. However, its composites exhibit different behavior. Although PEDOT has high ion resistance, the macroporous region around carbon particles, when included, allows ions to travel long distances with relatively high ion mobility. The distribution of electronic phases further reduces effective ion resistance [45]. Regarding PSS, acting as a connecting substance between the other two components, it swells in a water environment due to water and ion penetration, such as Na^+^, Cl^−^, and K^+^, reaching the internal structure of the hydrophilic polymer. Therefore, the nanocomposite PEDOT:PSS/Graphene material is a complex system where various physical processes contribute to charge transport. It can be assumed that within the interior of the PEDOT-based system, there are two main types of phase boundaries: PEDOT/PSS and PSS/Graphene. Studies of PEDOT:PSS and PEDOT:PSS/Graphene systems have shown that graphene and PEDOT + PSS agglomerates are located in the PSS matrix [32,47,48,49]. This suggests that the composite layer/electrolyte boundary can be simplified to PSS/electrolyte, as the other phases mainly contact the electrolyte through the PSS phase. Our EIS results show that the electrode material capacitive binds to the surrounding electrolyte through the formation of a double electric layer, in line with other studies [2,15,17,20]. This suggests hindered charge exchange at this boundary and initial chemical stability. From this perspective, PEDOT:PSS/Graphene seems suitable for manufacturing printed electrodes for capacitive sensors.

The process of wetting and destruction of the PSS phase is accelerated by electrochemical cycling as a result of potential-stimulated continuous absorption and release of electrolyte ions and associated volume changes [15,17]. Ion transfer is associated with expansion in receiving zones and contraction on the donating side [45]. Thus, structural changes occur in the layer during electrochemical loading, expressed in the penetration of ions from the electrolyte into the depth of the layer [2]. The presented CV dependencies in Appendix A show that the reduction in the hysteresis area (voltametric charge) is primarily the result of a decrease in the anodic slope at potentials above 0.3 V. Therefore, positive polarization, which allows for oxidative processes, can be considered as more destructively acting electrochemical loading. It is known that in an oxidative environment (such as H_2_O_2_), the thiophene sulfur atoms in PEDOT are over-oxidized and/or the double bonds in the conjugated structure are broken, leading to a reduced conjugation length and, therefore, reduced conductivity. As a result, during electrochemical aging or contact with an oxidative environment, the electrode becomes more insulating due to an increase in its impedance [23,49].

The significant change in the impedance characteristics during prolonged contact of the layer with the artificial sweat is probably caused by structural and chemical changes in the PEDOT:PSS/Graphene composite. As mentioned above, the changes occurring within three days result in a decrease in the capacitance of the double electric layer and an increase in the layer resistance by over 3 orders of magnitude. This indicates that homogeneity within the film increases during exposure to the electrolyte, associated with changes in thickness, ion adsorption on the film, and interior pores filled with electrolyte [40]. Over time, the distribution of the phase with the highest ion conductivity becomes uneven, decreasing and exposing PEDOT and graphene, while resistance values reach levels equivalent to layer breakage. Therefore, the results of aging in the electrolyte clearly show that PSS should be treated as water-soluble under operational conditions for wearable biosensors, i.e., in contact with sweat and periodic electrochemical loading. During this aging process, surface PSS molecules gradually detach from the composite, and after a day or two, exposure of the other two phases begins. Electrolyte access to the internal interphase boundaries of graphene-PSS leads to the breakdown of interphase bonds and delamination [20]. With time, the electrolyte gains access to deeper structures, reaching the boundary with the insulating substrate, resulting in the breakdown of the electrode. The presented EIS results show that the duration of use of PEDOT:PSS/Graphene layers as a conductive pattern material for monitoring electrochemical parameters in an electrolyte environment must be carefully determined. For longer tests, it is necessary to isolate the layers both from the atmosphere and from the electrolyte.

The perspectives for development and further investigations following these findings are multifaceted. Firstly, the development of additional protective measures or encapsulation techniques to improve the stability of PEDOT:PSS/Graphene layers in saline-acidic electrolytes could be explored. This would involve studying advanced coating materials or barrier technologies to mitigate the observed degradation during prolonged exposure to artificial sweat. Furthermore, it would be valuable to investigate a variety of additional dopants to the existing PEDOT:PSS/Graphene composition that could enhance long-term stability and performance. Examples of advanced coating materials could involve resilient polymer blends, engineered nanocomposites, or functionalized nanostructures with specific protective properties. As for barrier technologies, these could encompass specialized encapsulation methods, such as conformal coatings, or nanostructured encapsulants engineered to shield the conductive polymer layers from the corrosive effects of the electrolyte. These barrier technologies may be designed to prevent moisture penetration, block ion diffusion, and minimize chemical interactions that contribute to the observed degradation of the PEDOT:PSS/Graphene layers in artificial sweat. 

## 5. Conclusions

In this study, we investigated the electrochemically induced changes in PEDOT:PSS/Graphene ink layer on a flexible insulating PET substrate to demonstrate the potential for use of this material as printed conductive patterns and electrodes. The presented results show that the layer exhibits predominantly capacitive behavior in the range of −0.2 to 0.6 V vs. Ag/AgCl, with a cut-off frequency (*f_cut-off_*) of approximately 1 kHz, making it suitable for high-frequency impedance measurements (as Electrical Impedance Plethysmography and Impedance Cardiography) and as capacitive sensors in the low-frequency range. During electrochemical cycling at 1 V potential window over the first 10 cycles, the capacitance decreases by approximately 10%, stabilizing at approximately 90% of the initial value by the end of the 500-cycle test. Aging for 6 months under ambient conditions leads to a slight increase in impedance, demonstrating the material’s potential for storage under undemanding conditions.

The layer shows the most pronounced degradation during prolonged exposure to artificial sweat. The impedance of the layer increases due to an increase in charge transfer resistance (from 101 Ω to 1.1 MΩ) and internal resistance (from 30.6 MΩ to 5.5 × 10^15^ MΩ), indicating severe damage and destruction of the polymer layer, as confirmed by SEM observations. The presented results for aging in artificial sweat question the possibility of using spray-coated PEDOT:PSS/Graphene layers as the conductive pattern material for long-term monitoring of electrochemical parameters in a weakly acidic electrolytic environment. For longer tests, it is advisable to use additional protection to enhance its stability in saline-acidic electrolytes.

Understanding the stability and performance of PEDOT:PSS/Graphene in various conditions, such as exposure to artificial sweat and air, is crucial for determining its suitability for biosensor applications. Additionally, the examination of morphological changes and the ability to mitigate potential issues related to electrode–substrate interactions can contribute to the development of more effective and reliable biosensor technologies. Therefore, the findings of this study could have significant implications for the design and development of biosensors utilizing conductive polymer patterns.

Future research should focus on enhancing the stability of PEDOT:PSS/Graphene layers in saline electrolytes by exploring advanced protective coatings and barrier technologies. Investigating new dopants to improve long-term stability is also important. Barrier technologies like specialized encapsulation methods can prevent degradation in artificial sweat by shielding the layers from electrolyte corrosion and moisture penetration.

## Figures and Tables

**Figure 1 polymers-16-01706-f001:**
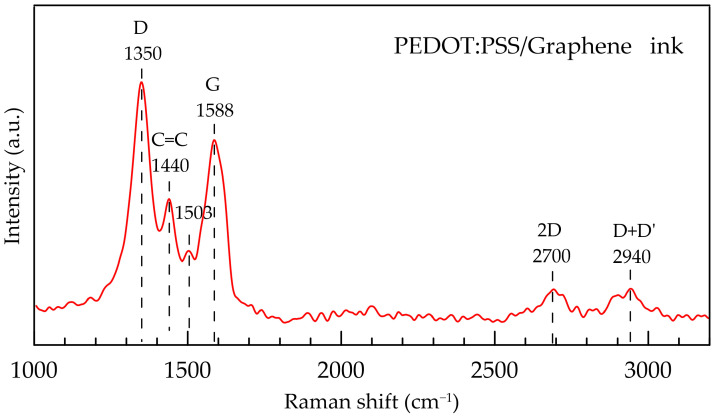
Raman spectra of PEDOT:PSS/Graphene ink.

**Figure 2 polymers-16-01706-f002:**
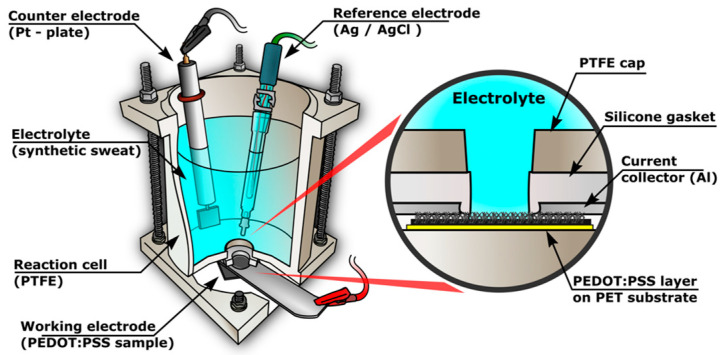
Schematic overview and cross-section of the electrochemical cell setup employed in the experiments with PEDOT:PSS electrodes in the current work.

**Figure 3 polymers-16-01706-f003:**
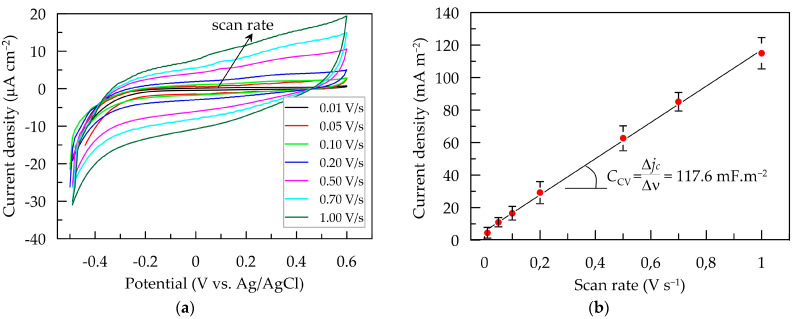
Scan rate influence on polarization response of PEDOT:PSS/Graphene layer in artificial sweat. (**a**) CV dependences at scan rate from 0.01 to 1.0 V cm^−2^; (**b**) dependence of the capacitive voltametric current on scan rate.

**Figure 4 polymers-16-01706-f004:**
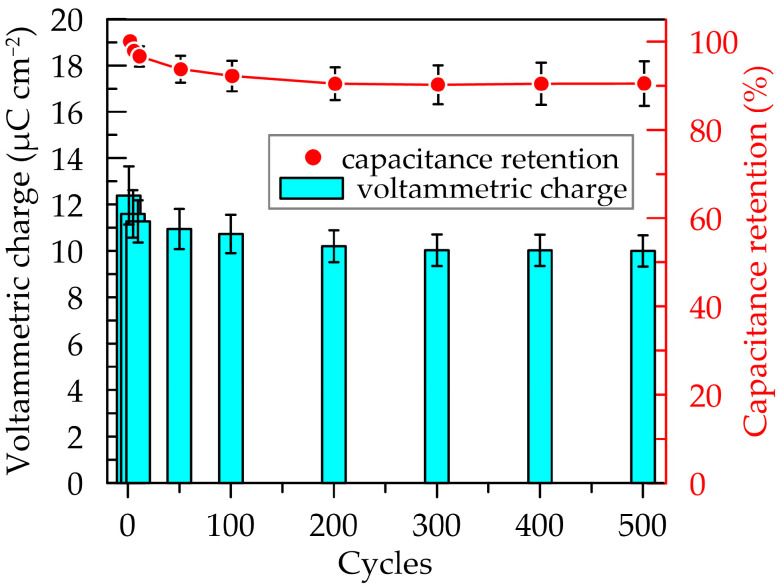
Voltammetry charge and capacitance retention dependencies upon 500 CV cycles. Measurements were performed between −0.3 and 0.7 V vs. Ag/AgCl at a scan rate of 0.1 V s^−1^.

**Figure 5 polymers-16-01706-f005:**
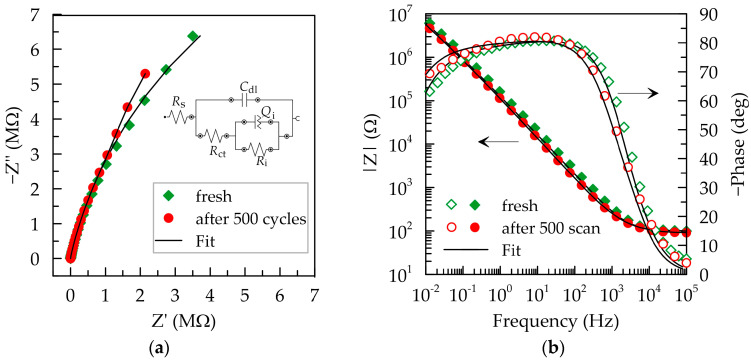
EIS results for the PEDOT:PSS/Graphene layer before and after 500 cycles at 100 mV/s. (**a**) Nyquist plot (inset: the equivalent circuit used to simulate the Nyquist plots); (**b**) Bode plot of magnitude and phase of the impedance.

**Figure 6 polymers-16-01706-f006:**
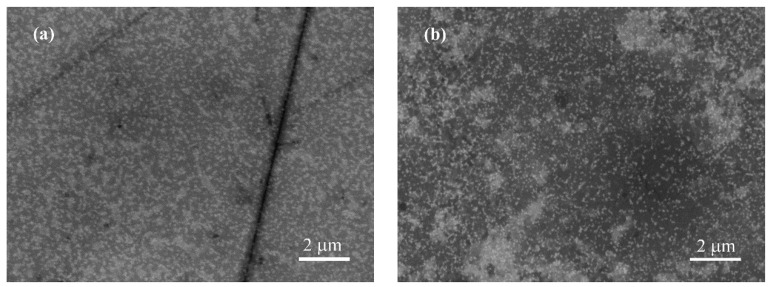
SEM imagines in the BSE mode of PEDOT:PSS/Graphene layer. (**a**) Before electrochemical testing; (**b**) after 500 scans in artificial sweat.

**Figure 7 polymers-16-01706-f007:**
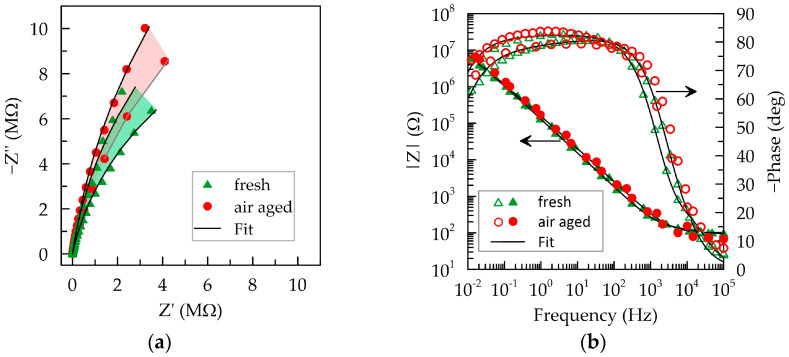
EIS dependencies in artificial sweat for fresh deposited (green triangles) composite PEDOT:PSS/Graphene layers and after 6 months (red circles) of exposure to the air. (**a**) Nyquist plots; (**b**) Bode plots.

**Figure 8 polymers-16-01706-f008:**
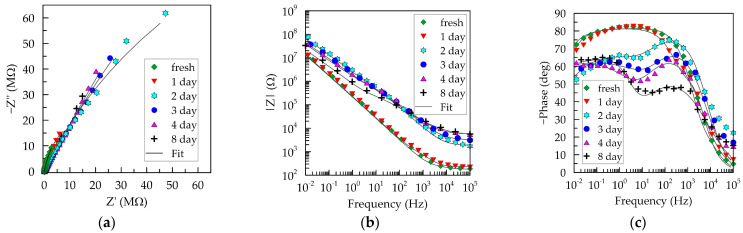
Aging of PEDOT:PSS/Graphene layer in artificial sweat. (**a**) Nyquist plot; (**b**) frequency dependence of impedance magnitude; (**c**) phase shift from frequency.

**Figure 9 polymers-16-01706-f009:**
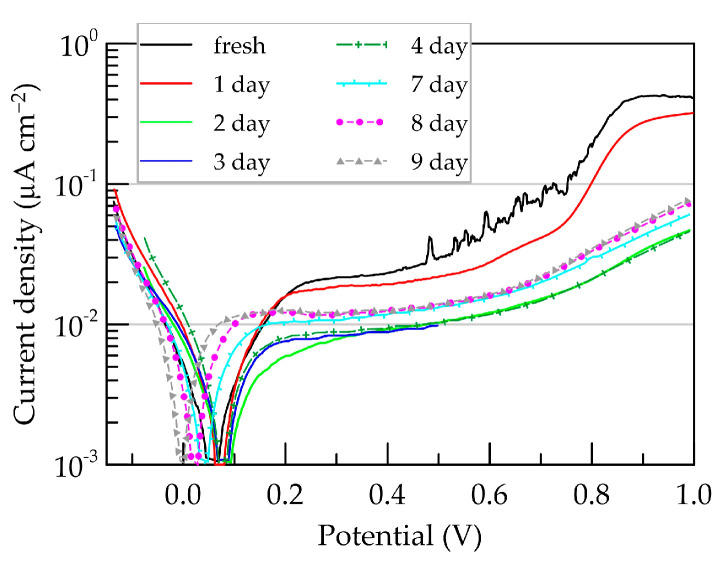
Polarization dependencies of PEDOT:PSS/Graphene at different soaking durations in artificial sweat.

**Figure 10 polymers-16-01706-f010:**
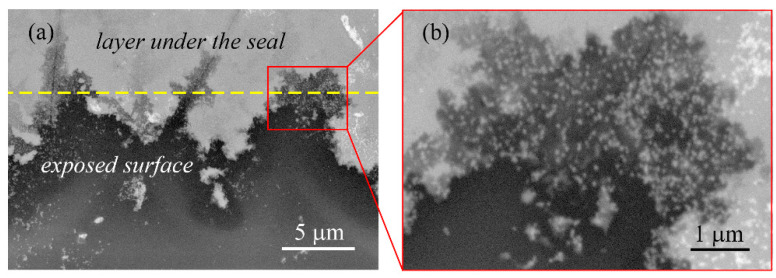
SEM imagines in the BSE mode of the PEDOT:PSS/Graphene layer after 9 days in artificial sweat and polarization tests at low magnification (**a**) and at high magnification (**b**).

**Table 1 polymers-16-01706-t001:** EIS data for PEDOT:PSS/Graphene of fresh layers and after aging through electrochemical cycling in artificial sweat and exposure to air.

Sample	*f*_cut-off_, kHz	*R*_s_, Ω	*R*_ct_, Ω	*R*_i_, MΩ	*Q*_i_, μS s^n^	*n*	*C*_dl_, nF	*χ* ^2^
fresh	2.53 ± 0.50	97.1 ± 3.2	100.6 ± 21.4	30.6 ± 5.2	1.16 ± 0.12	0.87 ± 0.03	257.6 ± 42.5	0.0005 ± 0.0002
after 500 cycles	1.63 ± 0.11	100.1 ± 4.3	207.7 ± 45.9	36.7 ± 11.6	1.34 ± 0.18	0.83 ± 0.01	450.6 ± 60.1	0.0014 ± 0.0006
aged in air	1.94 ± 0.59	120.4 ± 9.1	115.1 ± 33.7	40.6 ± 7.4	1.19 ± 0.22	0.86 ± 0.01	269.3 ± 34.3	0.0002 ± 0.0001

**Table 2 polymers-16-01706-t002:** Data from electrochemical impedance spectra of PEDOT:PSS/Graphene layers after aging in artificial sweat.

Soaking, Days	*f* _cut-off_	*R* _s_	*R* _ct_	*R* _i_	*Q*_i_,	*n* _i_	*Q* _dl_	*n* _dl_	*χ^2^*
	kHz	Ω	kΩ	MΩ	μS s^n^		nS s^n^		
0	2.34	97.10	0.101	30.60	1.160	0.870	257.6	1.00	0.0005
1	2.57	231.5	1.522	65.03	0.448	0.875	236.0	0.96	0.0023
2	5.89	1885	967.0	329.5	0.053	0.646	38.03	0.87	0.0124
3	4.17	3165	648.7	791.9	0.082	0.649	0.049	0.81	0.0045
4	2.19	5604	922.5	1271	0.101	0.671	0.056	0.77	0.0051
7	2.40	5222	1077	4.01 × 10^12^	0.052	0.843	0.153	0.64	0.0026
8	1.35	5160	547.0	5.469 × 10^15^	0.093	0.894	0.188	0.64	0.0037
9	0.76	4668	220.0	1.89 × 10^15^	0.121	0.801	0.209	0.67	0.0029

**Table 3 polymers-16-01706-t003:** Electrochemical parameters extracted from the polarization dependencies of the PEDOT:PSS/Graphene layer aging in artificial sweat.

Soaking, Days	*E*, V	*j*_0_, nA cm^−2^	*R*_p_, MΩ	Cathodic Tafel, V dec^−1^	Anodic Tafel, V dec^−1^
0	0.055	21.11	146.2	0.154	0.132
1	0.071	3.38	106.6	0.179	0.155
2	0.082	2.21	189.2	0.252	0.156
3	0.081	3.42	149.6	0.290	0.198
4	0.078	3.66	121.5	0.308	0.153
7	0.041	3.38	114.5	0.212	0.154
8	0.024	3.65	88.04	0.171	0.131
9	−0.001	3.85	75.33	0.162	0.113

## Data Availability

The original contributions presented in the study are included in the article/Appendix A, further inquiries can be directed to the corresponding author.

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
