# Peer review of "Electrochemical Investigation of PEDOT:PSS/Graphene Aging in Artificial Sweat"

_polymers, 2024, doi:10.3390/polym16121706_

Round 1

Reviewer 1 Report

Comments and Suggestions for Authors

In this research paper titled " Electrochemical investigation of PEDOT:PSS/Graphene aging in artificial sweat" the authors investigate the potential application of a composite consisting of PEDOT:PSS/Graphene, deposited via spray coating on a flexible substrate, as an autonomous conducting film for applications in wearable biosensor devices.

This paper is a followup work of the same same author, which has been published already in Tzaneva, B.; Aleksandrova, M.; Mateev, V.; Stefanov, B.; Iliev, I. Electrochemical Properties of PEDOT:PSS/Graphene Conductive Layers in Artificial Sweat. Sensors 2024, 24, 39. https://doi.org/10.3390/s24010039.

After completely reading the two manuscripts, it is unfortunate that I could not find novelty in,

1.     Motivation

2.     Material fabrication

3.     Investigation

4.     Characterization

Thus, publication of this work is not recommended at this time. Indeed, the authors conducted several studies in this paper, but when compared to previous work, this work is more akin to supplementary data or salami slicing. Suggest that the author work on new aspects of the same material and modify the manuscript before submitting it to any journal. 

Comments on the Quality of English Language

There are syntax errors 

Reviewer 2 Report

Comments and Suggestions for Authors

The authors reported the performance of PEDOT:PSS/Graphene on PET substrate aging in artificial sweat. The manuscript was well written with sufficient result data support the conclusion. The material was characterized by different characterization methods such as EIS, CV, LP, and SEM. However, there’re a few points the author should revised before the manuscript is published.

1)      More information is needed for the material characterization part. The sample preparation and SEM imaging conditions are needed, for example: accelerate voltage, vacuum mode?

2)      There is no proof for materials composition of the synthesized thin film on PET. The film was only characterized by KLA-Tencor Alpha Step 100. Raman is suggested for proving the graphene layer inside the composite.

3)      SEM image for figure 5 should be at higher magnification to show better comparison of the microstructure of the thin film. On the same thought, Figure 9 will show the exposed graphene sheet layer?

4)      Please double check the consistency of reference format, some ref has DOI and some don’t. 

Reviewer 3 Report

Comments and Suggestions for Authors

“Electrochemical investigation of PEDOT:PSS/Graphene aging in artificial sweat” is a research article with classical design, included introduction, material and methods, results. The article provide interesting result of changing material that could be applied in wearable biosensor for sweat analysis.

1. Figure 1. It’s very interesting construction of the cell. What is the reasons putting working electrode into the lower part of the electrochemical cell. Will changing of the electrode position effect of obtain date?

2. Figure 2. Please, add information how many tests were carried down for one measurement. Is it results from one electrode or from several electrodes with the same content. Add also CV curves that were used for the calculation.

3. line 213. Why do you suggest that faradaic reactions is not observed. There is no peak on the CV curve? Or this faradaic reactions is not limiting stage of electrochemical process?

4. line 216, Please, add the reference how was voltammetric capacitance calculated?

3. Figure 3. Please, add information how was capacitance retention estimated? Please, add also information how many tests were carried down for one measurement. Is it results from one electrode or from several electrodes with the same content. Add also CV curves.

4. The main question how will your biosensor work? What kind of biomaterials and electrochemical detector will be used? This information could be added into conclusion part.

5. line 476, You state “Prolonged contact of the layer with artificial sweat solution induces structural and chemical changes in the layer”. Could you add proposal chemical schema of this changing? And based on which method do you conclude it? There are a lot of microscope methods for structure study and electrochemical properties, but I couldn’t see methods for chemical structure study. Please, add more information at result, discussion and material and methods parts.

The manuscript are provide interesting results and can be published in a journal after some corrections and suggested improvements.

Round 2

Reviewer 1 Report

Comments and Suggestions for Authors

The authors improved the manuscript in the revised version. However, I have a few concerns.

1.     There is a conflict in the results between the previous and current work of the author.

The author’s previous work stated that “After 500 CV cycles in a potential window of 1 V (from −0.3 to 0.7 V), capacitance retention for most layers is around 94%, with minimal surface changes being observed in the layers.” [Tzaneva, B et al. Electrochemical Properties of PEDOT:PSS/Graphene Conductive Layers in Artificial Sweat. Sensors 2024, 24, 39. https://doi.org/10.3390/s24010039.]  and the current work stated that  “The results indicate that the layers exhibit predominant capacitive behavior in the potential range of −0.3 to 0.7 V vs. Ag/AgCl, with a cut-off frequency of about 1 kHz and retains 90% capacity after 500 cycles.” Since the author used the same synthesis procedure and the same composition. Why does the difference exist in the performance? This point should be addressed in the manuscript with proper citation of the previous work.

2.     In the materials section: The authors stated that “The hybrid material, comprising 1 mg mL−1 PEDOT:PSS/Graphene in dimethylformamide (DMF) solvent (Sigma Aldrich—Merck KGaA, Darmstadt, Germany) (0.2 mg 143 mL−1 PEDOT:PSS and 1 mg mL−1 electrochemically exfoliated graphene) was prepared.” The authors should provide a detailed procedure for how was this composite made. The synthesis procedure was newly reported (if so, need step-by-step preparation procedure) or following the previously reported one (if so, cite the paper) or directly purchased?

3.     In the materials section: The exact artificial sweat solution composition must be mentioned.

It is well known that artificial sweat consists of 22 mM urea, 5.5 mM lactic acid, 3 mM NH4+, 0.4 mM Ca2+, 50 μM Mg2+, 25 μM UA, 5 mM glucose, 10 mM K+ and 100 mM Na+. Artificial urine (AU) solution at pH 6.0 was prepared by dissolving 2.0 mM citric acid, 25 mM NaHCO3, 170 mM urea, 2.5 mM CaCl2, 90 mM NaCl, 2.0 mM MgSO4, 10 mM Na2SO4, 7 mM KH2PO4, 7 mM K2HPO4, and 25 mM NH4Cl in water. [Ref: (a) Y. Lei et al. Single-atom doping of MoS2 with manganese enables ultrasensitive detection of dopamine: Experimental and computational approach, Sci. Adv. 6 (2020) eabc4250. https://doi.org/10.1126/sciadv.abc4250.

(b)  J. Suriyaprakash et al. Immobilized Molecules’ Impact on the Efficacy of Nanocarbon Organic Sensors for Ultralow Dopamine Detection in Biofluids. Adv. Mater. Technol. 2022, 2200099. https://doi.org/10.1002/admt.202200099 ]. But the authors used only a few chemicals. Is it mistakenly left other chemicals (if so, mention the rest of the chemicals as listed above with proper citation) or author only use those reported chemicals to prepare artificial sweat (if so, the author should experiment with a standard artificial solution)? Because the work aims to investigate the aging properties of polymer composites in artificial sweat. This point is significant.

4.     In Fig. 2a and 2b, The scan rate and the current density value is not match. How can the current density reach 115 mA/cm2 when the CV curve of 1V/s maximum reaches the current density of 20 uA/cm? The same applies to other scan rate values as well. The author should pay attention to this. Is the data of Figure 3a in the present work and Figure 5d in the author’s previous work the same? If so author should not use it, try to present it in another way with proper citation.  

5.     In discussion: line  500-502, The authors state that “The significant change in the impedance characteristics during prolonged contact of  the layer with the artificial sweat is probably caused by structural and chemical changes in the PEDOT:PSS/Graphene composite.” The author only provides evidence for the morphology changes observed in the aged polymer composite using SEM analysis. It can be used to describe structural changes, but not chemical changes.  To prove the author's claim, additional analysis, such as FTIR and EDX analysis of the polymer before and after, as well as a discussion of the chemical changes, are required.

Round 3

Reviewer 1 Report

Comments and Suggestions for Authors

The Authors have addressed all of my concerns with the original manuscript. The revised manuscript is recommended for publication.